# Upper and Lower Respiratory Signs and Symptoms in Workers Occupationally Exposed to Flour Dust

**DOI:** 10.3390/ijerph17197075

**Published:** 2020-09-27

**Authors:** Maria Angiola Crivellaro, Giancarlo Ottaviano, Pietro Maculan, Alfonso Luca Pendolino, Liviano Vianello, Paola Mason, Francesco Gioffrè, Rosana Bizzotto, Bruno Scarpa, Edi Simoni, Laura Astolfi, Piero Maestrelli, Maria Luisa Scapellato, Mariella Carrieri, Andrea Trevisan

**Affiliations:** 1Department of Cardiac Thoracic Vascular Sciences and Public Health, University of Padova, 35128 Padova, Italy; mariaangiola.crivellaro@aopd.it (M.A.C.); pietro.maculan@unipd.it (P.M.); paola.mason.1@unipd.it (P.M.); piero.maestrelli@unipd.it (P.M.); marialuisa.scapellato@unipd.it (M.L.S.); mariella.carrieri@unipd.it (M.C.); 2Department of Neurosciences, University of Padova, 35128 Padova, Italy; giancarlo.ottaviano@unipd.it (G.O.); edi.simoni@unipd.it (E.S.); laura.astolfi@unipd.it (L.A.); 3Department of ENT, Royal National Throat, Nose and Ear Hospital, London WC1X 8DA, UK; alfonso.pendolino@nhs.net; 4SPISAL, Azienda ULSS7 Pedemontana, 36061 Bassano del Grappa (VI), Italy; liviano.vianello@aulss7.veneto.it; 5SPISAL, Azienda ULSS6 Euganea, 35128 Padova, Italy; francesco.gioffre@aulss6.veneto.it (F.G.); rosana.bizzotto@aulss6.veneto.it (R.B.); 6Department of Statistical Sciences, University of Padova, 35128 Padova, Italybruno.scarpa@unipd.it; 7Department of Mathematics “Tullio Levi-Civita”, University of Padova, 35128 Padova, Italy

**Keywords:** flour dust, occupational exposure, PNIF, Tiffeneau index, smell, skin prick test, nasal cytology

## Abstract

A group of 142 bakers was studied in order to investigate the relationship between higher/lower respiratory signs/symptoms and inflammation biomarkers and occupational exposure to flour dust. A complete upper and lower respiratory tract evaluation was performed. Seven percent of bakers complained of lower respiratory symptoms, while 22% of them complained of upper respiratory symptoms. Fifty five percent of the bakers were allergic, and 37.1% showed sensitization to occupational allergens. Abnormal spirometries were found in 15% of bakers, while fractional exhaled nitric oxide (FeNO) was above the normal reference in 24.5% of them. Moreover, 23.8% of bakers were found to be hyposmic. Population mean peak nasal inspiratory flow (PNIF) was in the normal range even if almost all the workers suffered from neutrophilic rhinitis at nasal cytology with the number of nasal neutrophils increasing with the increase of the duration of exposure to flour dust (*p* = 0.03). PNIF and FEV1 (forced expiratory volume in the 1st second) showed a positive correlation (*p* = 0.03; r = 0.19). The Tiffeneau index decreased with the increase of dust (*p* = 0.017). A similar result was obtained once we divided our population into smokers and non-smokers (*p* = 0.021). Long-term exposure to bakery dusts can lead to a status of minimal nasal inflammation and allergy.

## 1. Introduction

Bakery workers, including confectioners, flour millers, food processors, and supermarket bakers, with a work-related daily exposure to flour dust may develop over time an allergic sensitization to high-molecular-weight flour allergens, which can finally lead to an allergic disease (i.e., rhinitis, asthma, conjunctivitis, or dermatitis). An elevated risk of sensitization has also been shown to be present even at relatively low exposure levels [1]. As a confirmation of this, bakers have frequently been found to be atopic at skin or immunoglobulin E (IgE) tests, with approximately 60% of bakery workers with respiratory symptoms showing specific IgE responses to the inhalation of flour allergens [2,3].

Baker’s asthma is one of commonest causes of occupational asthma with a reported annual incidence ranging from 1 to 10 cases per 1000 bakery workers [4]. The main agents that cause baker’s asthma are cereal flours (wheat, rye, barley), enzymes, soy and buckwheat, and other allergens, such as molds, yeast, eggs, milk powder, sesame seeds, and nuts [5]. In a previous longitudinal Italian study, the sensitization to flour or α-amylase (an enzyme added to flour to help increase the volume of bread preparation) was found to be a significant predictor (odds ratio 4.3) for work-related respiratory symptoms [6]. Moreover, a cross-sectional study conducted on supermarket bakery workers demonstrated that the increase of fractional exhaled nitric oxide (FeNO) is strongly associated with wheat IgE level, work-related rhinitis symptoms, and asthma, independently from the smoking or atopy status [7]. Conversely, storage mites are no longer considered bakery allergens since sensitization rates among bakers are similar to those reported in the general population [8].

Respiratory symptoms could develop months, years, or even decades after a continuous exposure to flour dust and may reduce or resolve completely during the time off from the bakery [9]. In general, symptoms of allergic occupational rhinitis precede occupational asthma onset [3].

Published studies on exposure–response relationship are exiguous. The available ones all showed a positive relation, namely a higher prevalence of chest and/or nasal symptoms in response to a higher exposure to bakery dust. (Recommendation from the Scientific Committee on Occupational Exposure Limits for Flour Dust, SCOEL/SUM/123 December 2008). Similar results have also been reported for sensitization to flour, usually wheat [8,10].

In the present study, skin prick tests (SPTs) for common airborne, flours and storage mite allergens, spirometry, FeNO, nasal cytology, peak nasal inspiratory flow (PNIF), the Screening 12 Test, along with the Sino-Nasal Outcome Test (SNOT-22) questionnaire, and a Visual Analog Scale (VAS) for the main chronic rhinosinusitis symptoms were performed in a group of bakery workers referred to a specific medical surveillance program in order to investigate whether occupational exposure to flour dust could influence (1) higher/lower respiratory signs/symptoms and (2) airway inflammation biomarkers.

## 2. Materials and Methods

### 2.1. Population

Artisan (family businesses with a limited number of workers) and industrial (with a more complex, structured organizational model) bakers were recruited randomly (among small, medium, and large facilities) from 26 bakeries in the province of Padua (also randomly chosen from 946 bakeries), North-Eastern Italy, and participated in this surveillance program study. The study was conducted from February 2017 to May 2018. A detailed working, medical and medication history, exploring work-related exposure to allergens, duration of exposure to flours and general and flour-related atopies known both in the participants and their families, common signs and symptoms of allergic disease of the upper and lower airways, and steroid and/or antihistaminic drug use, was taken.

### 2.2. Ethical Considerations

The present investigation was a prospective study conducted in accordance with the 1996 Helsinki Declaration. All subjects gave their written informed consent for their inclusion in the study and for the clinical publication of the data. Data were examined in agreement with the Italian privacy and sensible data laws (D.Lgs 196/03) and the internal regulation of the Sections involved. The study was approved by the Hospital Ethical Committee (4136/U6/17).

### 2.3. Allergy Evaluation

Skin prick test (SPT) was performed using “single puncture” lancets (Allergopharma-Rome, Italy), according to Bousquet et al. [3] and Heinzerling et al. [11].

### 2.4. Lower Airway Examination

The Asthma Control Test (ACT) was used to evaluate lower respiratory symptoms. Spirometry was performed following the guidelines of the American Thoracic Society/European Respiratory Society (ATS/ERS) standards [12]. A Fukuda Sangyo SpiroAnalyzer SY-150 portable spirometer was used. The normal predicted values were according to European Coal and Steel Community 1971 (ECSC71). Spirometric results were categorized according to the literature [13]. Evaluation of FeNO was done using NObreath^®^ (Bedfont Scientific Ltd.—Kent, England), a portable electrochemical analyzer. Participants underwent the measurement according to the literature [14]. NObreath^®^ is calibrated once a year as part of a regular maintenance service provided by the manufacturer.

### 2.5. Nasal Examination

All subjects were evaluated for nasal symptoms and sino-nasal quality of life using the SNOT-22 questionnaire and the VAS for nasal obstruction, rhinorrhea, post-nasal drip, facial pain, and olfactory dysfunction [15,16,17]. Nasal endoscopy was performed in all subjects using either a rigid, 0° or 30° endoscope to exclude the presence of nasal polyps [18]. A portable “Youlten” flowmeter (Clement Clark International—Harlow, England) was used for PNIF measurement as previously reported [19,20,21,22,23]. The Screening 12 Test ^®^, a quick suprathreshold validated smell identification test based on 12 odors, was also performed to investigate any olfactory dysfunction [24,25]. Nasal cytology was examined through anterior rhinoscopy using a Rhino-Probe (Arlington Scientific Inc., Springville, UT, USA) nasal curette, and all the specimens were examined under the light microscope by the same operator (G.O.). The cytological variables considered were the total number of ciliated cells, with and without hyperchromatic supranuclear stria (HSS), and the total number of inflammatory cells (neutrophil granulocytes and eosinophil granulocytes) counted for each specimen in five separate high-power fields (HPF, original magnification ×100) [26,27,28].

### 2.6. Dust Sampling

Industrial hygiene sampling sessions were performed in all bakeries by means of personal and area sampling [29]. For the dust analysis, we divided the measured personal dust concentration into three different levels chosen by considering the limit of 1 mg/m^3^ recommended by SCOEL, namely dust level (1) <1 mg/m^3^, dust level (2) between 1 and 2 mg/m^3^, and dust level (3) ≥2 mg/m^3^.

### 2.7. Statistical Analyses

We reported continuous variables as mean, standard deviation (SD), and range. Categorical variables were reported as number and frequency. To compare continuous variables between groups (smokers, dust exposure, etc.), we used the analysis of variance test. To verify the existence of correlation between the measured quantitative parameters, we obtained a regression line and obtained the slope and its *p*-value. Multiple regression models and additive models have been fitted to explain all the relevant variables, by selecting influencing variables via all subset selection, where possible, and via stepwise selection where the computational burden was too high. The level of significance was set at 0.05. Values in the range of 0.05 *p* < 0.15 were considered to indicate a statistical trend.

## 3. Results

One hundred and forty-two workers (98 men, 44 women) with a long-term exposure to flour dust because of their work-related environment were randomly sampled from artisan and industrial bakeries of the province of Padua. Table 1 shows the participants’ main characteristics.

### 3.1. Main Allergic and Lower Airway Signs and Symptoms

According to the ACT questionnaires, only 10 bakers (7%) had lower respiratory symptoms such as cough, wheezing, breathlessness, waking up during the night for tight chest, and 14 (9%) had a history of cutaneous diseases. Furthermore, 23% of the evaluated subjects reported a known sensitization to at least one allergen, whilst 29% reported atopy familiarity.

Table 2 shows the SPT results. Two subjects did not undergo SPT because they were on antihistamines at the moment of the test.

Seventy-seven (55%) bakers were positive to at least one of the allergens tested, while 52 (37.1%) had sensitization to occupational allergens, i.e., flours or flour mites. Bakers sensitized to cereal flours showed an average exposure time of 16.15 (SD = 10.2) years, while those not sensitized showed an average exposure time of 13.72 (SD = 12.2) years.

Two spirometries were not performed due to a concomitant inhalation therapy. Of the 140 spirometries performed, 7 were not validated due to a lack of compliance. In the remaining 133 spirometries, abnormal findings were detected in 20 (15%) of them: 9 subjects showed bronchial obstruction, whilst 11 had a reduction of pulmonary volumes. FeNO mean values were above the normal reference (23.7 ± 25.0, range 2–166) in 34 (24.5%) of the subjects. In addition, 22 workers who had FeNO above the normal values were also positive to at least one of the tested allergens at SPT. Table 3 shows the main lower respiratory results found in the population.

When we evaluated the relationships between flour, mites and environmental allergen SPT positivity, and pulmonary function (FEV1, FEV1%, Tiffeneau index, FeNO), we found a significant direct relationship between airborne allergy and FeNO (*r* = 0.2, *p* = 0.01), regardless of the effect of age, BMI, smoking status, duration of occupational exposure, or dust levels.

Finally, we investigated whether being positive to SPT for environmental allergens could be a risk factor for being positive to flour allergens. No significant relationships were found. In addition, we did not find any effect of smoking or demographic characteristics on flour allergy.

### 3.2. Main Upper Airway Signs and Symptoms

Twelve subjects refused the nasal evaluation because of poor compliance. Of the remaining 130 workers, 29 (22.31%) complained of upper respiratory symptoms, based on SNOT-22 results [30].

A similar percentage of subjects showed to be hyposmic (23.8%) at the olfactory identification test. Of the 130 subjects who completed the nasal evaluation, nasal cytology was not available in 9 subjects because their slides were not readable due to the small amount of mucous collected. Table 4 shows the main rhinological characteristics of the population.

### 3.3. Bakery Dust Sampling

Flour dust levels were highly variable in different workplaces, ranging from 0.097 to 14.005 mg/m^3^ (Figure 1).

### 3.4. Relationship between Upper and Lower Respiratory Function

FEV1 and PNIF showed (Figure 2) a significant positive correlation (*p* = 0.03; r = 0.19).

### 3.5. The Effects of Smoke on Airways

The Tiffeneau index (FEV1/FVC) was significantly lower in the smokers when compared to that in the non-smokers (*p* = 0.0007; means difference = −4.25) (Figure 3).

Apart from the SNOT-22 and the VAS for nasal obstruction scores, which showed a significant increase in the smokers when compared to that in the non-smokers (*p* = 0.003, means difference = 6.8 and *p* = 0.02, means difference = 0.9, respectively) (Figure 4), no other significant differences were observed with regard to the other parameters evaluated (Table 5).

### 3.6. Relationship between Dust Exposure (Time and Concentration) and Lower Respiratory Investigation Results

The results of multiple regression models show that, among the lower respiratory findings, the only parameter influenced by dust concentration was the Tiffeneau index. In particular, this index decreased (Figure 5) when dust concentrations increased over 2 mg/m^3^ (*p* = 0.017; means difference = −3.2).

A similar result was obtained once we divided our population into smokers and non-smokers. Specifically, the Tiffeneau index decreased with the increase of dust concentration over 2 mg/m^3^ in the group of smokers (*p* = 0.021; means difference = −4.2). Conversely, no correlation was found in the non-smoker group.

Considering the time of exposure, we found a significant correlation between the Tiffeneau index and the number of years of exposure to dust (*p* = 0.02, *r* = −0.20). Again, multiple regression and consequent variable selection show that the only significant relationship is the decreasing of Tiffeneau index (Figure 5) with the increase of the years of exposure to dust in the group of smokers (*p* = 0.002, *r* = −0.34).

FEV1, FEV1%, and FeNO did not change with dust concentrations, smoke exposure, or time of exposure (Table 5). On considering logarithms of exposure time, no significant differences in the results were observed.

### 3.7. Relationship between Dust Exposure (Time and Concentration) and Upper Respiratory Investigation Results

Overall, we could not find any significant effect of dust on nasal parameters. However, the number of neutrophils at nasal cytology significantly increased (Figure 6) with the increase of the exposure to dust in terms of years (*p* = 0.03, *r* = 0.15).

Furthermore, once we divided the population according to the different classes of dust exposure, we observed a marginal increase of VAS for nasal obstruction with the increase of dust concentrations in the non-smokers (*p* = 0.11, *r* = −0.14).

Again, on considering the logarithms of exposure time, no significant differences in the results were observed.

## 4. Discussion

In the present study, the analysis of the sensitization pattern of the bakers showed a 55% rate of specific sensitization to work-related aeroallergens (storage mites and flours). Additionally, the prevalence of atopy in our population was higher than that reported in the general population (about 20%) [31,32], suggesting that exposure to flour dust may increases the risk of developing atopy over time. Nevertheless, an accurate identification of the inciting agents would have required a specific inhalation challenge, which, however, must be carried out in a hospital setting [33]. Our data also suggest that such sensitization could have been acquired during the work, and this is supported by the fact that bakers sensitized to cereal flours showed on average a longer exposure time (mean exposure time = 16.15 ± 10.2) to dust than those not sensitized (mean exposure time = 13.72 ± 12.2).

In the case of subjects co-sensitized to potentially cross-reactive aeroallergens, equal to the remaining 47% of all bakers with allergic sensitization, cross-inhibition experiments could have suggested the source of primary or prevalent sensitization. In fact, a high positivity rate of specific provocation tests using cereal flour has already been shown in subjects not professionally exposed to flour dust but with a sensitivity to grass pollen [34]. Our results also corroborate previous findings, which reported the possibility of sensitization to specific aeroallergens of minor mites and cereal flours [35,36,37].

In spite of the high number of allergic subjects found at the SPT (55%), a very low number of bakers complained of respiratory symptoms, which can be partly justified by the long and persistent exposure to the allergens [38], or by an underestimation of the reported symptoms. However, our population of bakers complained more of upper than of lower respiratory symptoms. A condition of nasal mucosa inflammation was also confirmed at the nasal cytology, whereas the majority of the subjects (86.6%) were found to have a neutrophilic rhinitis. Taking into account that more than 50% of our population was found to be positive to at least one of the tested allergens, we would have expected to find a higher expression of eosinophils at nasal cytology. However, it has been reported that a long exposure to dust (in our population, the mean amount of working years in a bakery was 14) can produce a status of minimal persistent inflammation, which is characterized, at nasal cytology, by a high presence of neutrophils and a very low ciliated cell expression [28,39,40]. In this regard, 45.5% of the bakers did not present ciliated cells at nasal cytology. Interestingly, non-allergic subjects also showed a similar cytological pattern, which could be secondary to a non-allergic (“cellular”) vasomotor rhinitis due to the persistent contact with a dusty environment [41]. Additionally, an olfactory dysfunction was observed in about 24% of the bakers enrolled. Of note, none of them had nasal polyps at nasal endoscopy, excluding any mechanical blockage of the olfactory cleft; thus, the lower olfactory function could most probably be secondary to a long-standing nasal mucosal inflammation.

The apparent contrast between the high percentage of atopic and rhinitis subjects and the low percentage of symptoms reported could partially be justified by the fact that the majority of the bakers enrolled were artisans and were either the owners of the company or owners’ relatives/friends. In particular, most of the bakeries considered for the study were family-run businesses, with a median number of five bakers each. Given this background, it is possible that a high percentage of the subjects involved will have voluntarily underestimated their symptoms in order to protect their job position or the company they were working for.

With regard to the functional respiratory tests, only 15% of the bakers (20 subjects) were found to have a pathological pulmonary function at the spirometry. FeNO was above the normal range in a higher percentage of subjects (about 25%), confirming it to be a good marker of pulmonary status in response to allergens. In fact, in a previous study, a continued exposure to occupational airborne allergens had been identified as the main determinants of longitudinal changes in FeNO [42]. In general, the mean PNIF value (163.7 L/min) was on the normal range for adults [19,43]. Of note, PNIF values were not significantly different between smokers and non-smokers, even though, as expected, the formers showed a significantly higher SNOT-22 and VAS for nasal obstruction scores and a significantly lower Tiffanau index. Once we analyzed the correlations between the upper and lower respiratory objective and subjective parameters and the exposure to dust and smoke, although a significant correlation was found between FEV1 and PNIF values (and, as expected, both values correlated with sex, height, BMI, and age), PNIF did not correlate with any of the variables considered. In particular, no correlation was found with dust exposure, either in terms of concentration or time of exposure. On the contrary, the Tiffeneau index (FEV1/FVC) negatively correlated with both the dust concentration found in the workplaces and the time of exposure in years. Interestingly, once we separated the smokers from the non-smokers, this correlation remained significant, even if marginally, only in the group of smokers meaning that there could be an overlapping effect of the smoking status and the exposure to dust on the chest function, with the Tiffeneau index significantly decreasing at the increase of both the dust concentration (*p* < 0.01) and the number of years of exposure to dust. Although the objective nasal parameters studied (i.e., nasal cytology and PNIF) did not correlate either with the dust concentration or with the time of exposure to dust, it should be highlighted that the VAS for nasal obstruction was shown to increase significantly with the increase of dust concentration exposure in the non-smoker bakers, keeping with the nasal minimal persistent inflammation observed at nasal cytology.

The main strength of the study was that a large number of subjects, living in the same geographical area, with a long-standing work-related exposure to the same allergens were enrolled and underwent a complete evaluation of the upper and lower airways. The prospective design of the study and the relatively short period of time in which it was conducted constitute other important strengths. In particular, the peculiarity of the study is based on the fact that it was carried out directly in the workplace, so it has to be considered among the so-called real-life studies.

The main study limitation is secondary to the nature of the companies in Italy, especially those in the North East of the country. In this area, the majority of them are small and mainly family-run. This could have influenced the results of the reported symptoms. Moreover, considering that both spirometry and PNIF values were within the normal range in the majority of the subjects, it could be argued that, at least in our population, bakery dust had a very limited effect on the respiratory system and that most of the bakers will develop only mild or no respiratory symptoms after a work-related exposure to bakery allergens. On the other hand, more than 50% of the subjects were allergic and the majority of them (86.6%) had neutrophilic rhinitis, meaning that long-term exposure to bakery dust can trigger a nasal inflammation. Since occupational pathology usually develops in the first periods of work [44,45], it is also possible that allergic workers with sensitization to occupational allergens may have already changed type of work, so our results could be biased by a healthy worker effect. Our estimates are lower than expected and likely lower than it could be without this selection bias.

## 5. Conclusions

Our results suggest that a long-term exposure to work-related bakery dusts can lead to (1) development of sensitization to occupational allergens (37.1%), (2) a status of minimal nasal inflammation, and (3) lower respiratory airway inflammation.

More than half of the bakers were positive to at least one of the tested allergens, while one-third were positive to occupational allergens at SPT. Long-term exposure to flour dust can create a status of minimal persistent inflammation in the nose, which manifests, at nasal cytology, with a neutrophilic rhinitis. The Tiffeneau index was one of the few parameters studied that correlated with the dust concentration, suggesting that this index could be the best spirometry parameter to consider when evaluating the pulmonary function in this category of workers. Additionally, FeNO appeared to be useful in identifying possible early respiratory changes.

Periodic surveillance tests, including spirometry, should be encouraged, so that the occupational physician can easily notice the onset of a lung function decline and suggest further investigations (e.g., bronchodilation tests, specific bronchostimulation challenge, sputum analysis, etc.). During the surveillance programs, occupational physicians should advice their patients to consider quitting smoking in order to avoid an uncontrolled decline in the lung function. In fact, in our populations, smoker bakers showed a worse respiratory outcome than non-smokers under the same number of years of exposure to dust.

More studies on this important and very attractive field are necessary to confirm the present results. An analysis of the occupational allergens by means of specific-IgE and new molecular diagnosis with specific recombinant allergens is also encouraged.

## Figures and Tables

**Figure 1 ijerph-17-07075-f001:**
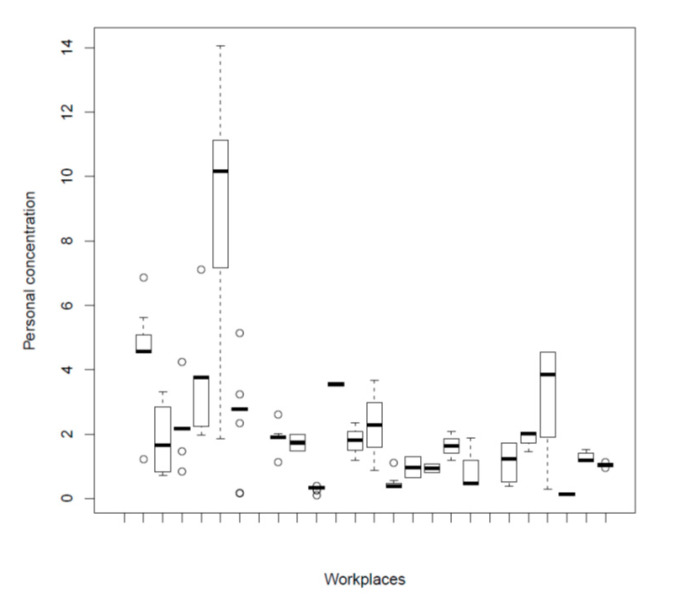
Flour dust levels in the different workplaces. Dust levels: mg/m^3^.

**Figure 2 ijerph-17-07075-f002:**
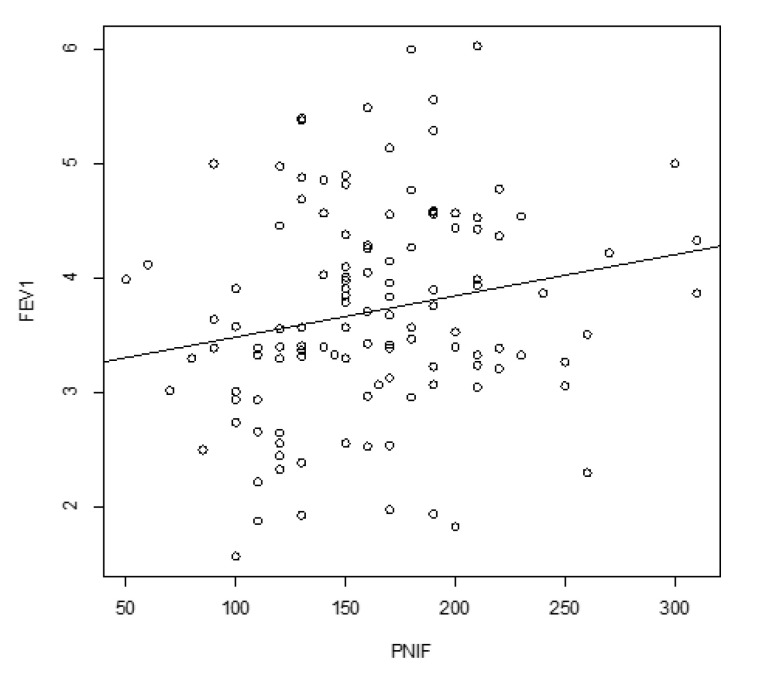
Correlation between FEV1 and PNIF. PNIF: peak nasal inspiratory flow; FEV1: forced expiratory volume in the 1st second.

**Figure 3 ijerph-17-07075-f003:**
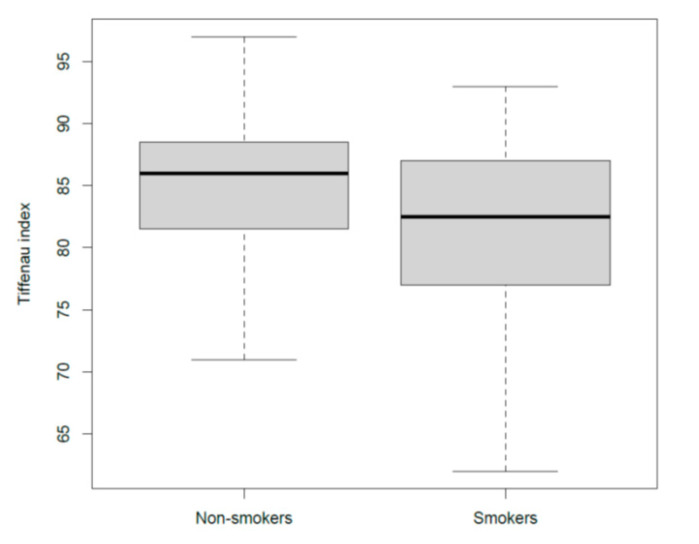
Tiffeneau index (FEV1/FVC) in smokers and non-smokers.

**Figure 4 ijerph-17-07075-f004:**
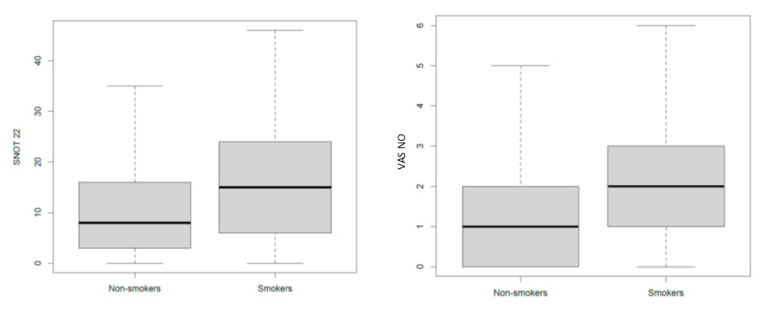
Nasal symptoms in smokers and non-smokers. SNOT-22: Sino-Nasal Outcome-22 Test; VAS NO: Visual Analog Scale for nasal obstruction.

**Figure 5 ijerph-17-07075-f005:**
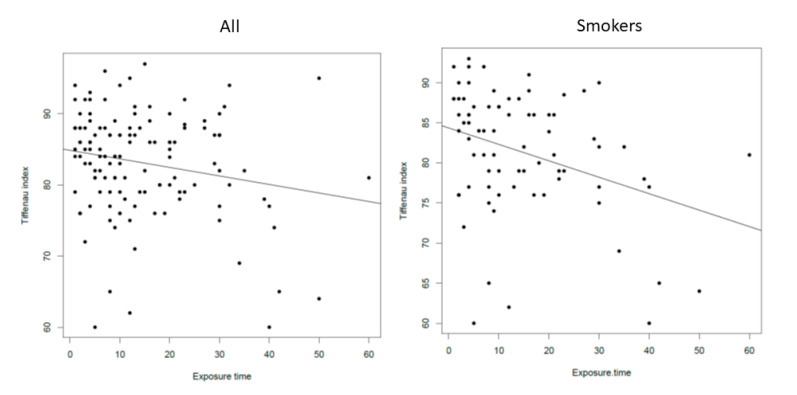
On the left, the correlation between Tiffeneau index (FEV1/FVC) and exposure time for all the bakers studied (*p* = 0.02, *r* = −0.20); on the right, the correlation between Tiffeneau index (FEV1/FVC) and exposure time for the bakers who were smokers (*p* = 0.002, *r* = −0.34).

**Figure 6 ijerph-17-07075-f006:**
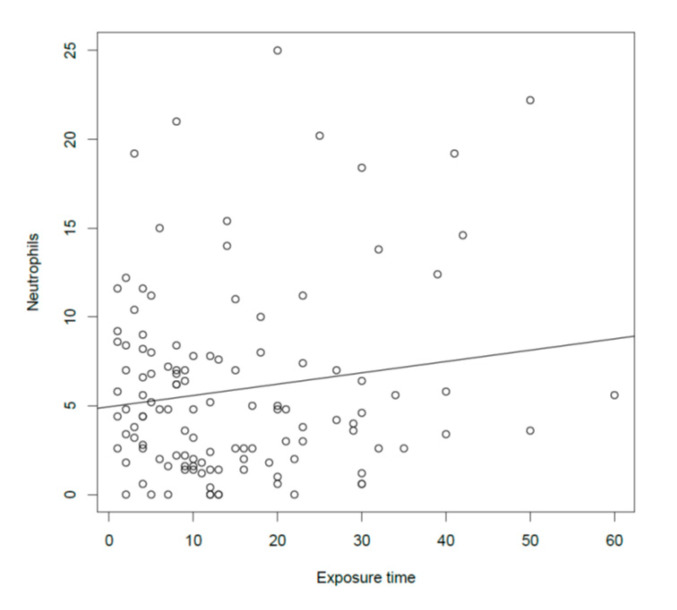
Correlation between the number of neutrophils at nasal cytology and the exposure time.

**Table 1 ijerph-17-07075-t001:** Main participants’ characteristics.

		Overall (N = 142)
Demographic Characteristics	N (%)	mean (SD)
Age (years)		39.9 (11.3)
Sex		
Male	98 (69.0)	39.9 (12.2)
Female	44 (31.0)	39.8 (9.3)
Height (cm)		175.4 (27.3)
Weight (kg)		77.3 (14.8)
Body Mass Index (kg/m^2^)		25.5 (4.3)
Occupational Factors		
Exposure time (years)		14.1 (11.8)
Personal dust concentration during working hours (mg/m^3^)		2.3 (2.3)
Smoking habit		
Smoker	81 (57.0)	39.0 (12.2)
Non-Smoker	61 (43.0)	41.1 (10.0)

**Table 2 ijerph-17-07075-t002:** Skin prick test results.

Part A		
Atopy status	Non-allergic	Allergic
	N (%)	N (%)
	63 (45.0)	77 (55.0)
Sensitization to occupational allergens	no	yes
	N (%)	N (%)
	88 (62.9)	52 (37.1)
**Part B**		
Allergens	Negatives	Positives
	N (%)	N (%)
**Airborne Allergens SPT**	75 (53.6)	65 (46.4)
*Dermatophagoides pteronyssinus*	91 (65.0)	49 (35.0)
*Dermatophagoides farinae*	98 (70.0)	42 (30.0)
*Alternaria*	139 (99.3)	1 (0.7)
Grass mix pollen	117 (83.6)	23 (16.4)
*Zea Mays* pollen	124 (88.6)	16 (11.4)
Birch pollen	128 (91.4)	12 (8.6)
Hazel pollen	132 (94.3)	8 (5.7)
*Artemisia* pollen	125 (89.3)	15 (10.7)
**Storage Mites Allergens SPT**	99 (70.7)	41 (29.3)
*Acarus siro*	121 (86.4)	19 (13.6)
*Glycyphagus domesticus*	112 (80.0)	28 (20.0)
*Tyrophagus putrescentiae*	112 (80.0)	28 (20.0)
*Lepidoglyphus destructor*	111 (79.3)	29 (20.7)
**Flour Allergens SPT**	114 (81.4)	26 (18.6)
Corn flour	120 (85.7)	20 (14.3)
Barley flour	134 (95.7)	6 (4.3)
Rye flour	135 (96.4)	5 (3.6)
Oats flour	135 (96.4)	5 (3.6)
Wheat flour	124 (88.6)	16 (11.4)
Whole-wheat flour	132 (94.3)	8 (5.7)
Soy flour	130 (92.9)	10 (7.1)
Yeast	136 (97.1)	4 (2.9)

**Table 3 ijerph-17-07075-t003:** Main lower respiratory results.

Respiratory Findings	Mean (SD)
ACT	24.2 (2.5)
Spirometry Parameters *	
FVC (L)	4.4 (1.3)
FVC% pred.	82.9 (24.5)
FEV1 (L)	3.6 (1.1)
FEV1% pred.	94.5 (54.7)
FEV1/FVC	83.1 (7.2)
FeNO (ppb) **	23.7 (25.0)

* Available for 133 subjects, ** fractional exhaled nitric oxide (FeNO) was gathered for 139 subjects, ACT: Asthma Control Test, FVC: forced vital capacity, L: liters, FEV1: forced expiratory volume in the 1st second, FEV1/FVC: Tiffeneau index.

**Table 4 ijerph-17-07075-t004:** Main upper respiratory results.

		Overall (N = 130)
Rhinological Findings	N (%)	Mean (SD)
PNIF (L/min)		163.7 (49.8)
Visual Analog Scale		
Nasal obstruction	2.0 (2.2)	
Rhinorrhea	1.3 (2.0)	
Post-nasal drip	1.0 (1.8)	
Frontal headache	0.3 (1.0)	
Olfaction reduction	0.9 (1.9)	
SNOT-22		14.4 (13.2)
Normal	98 (75.4)	
Pathologic	32 (24.6)	
Screening 12 Test		8.7 (1.9)
Hyposmic *	31 (23.8)	
Normosmic *	99 (76.2)	
Nasal Cytology Findings **		
Neutrophils (found in 105 subjects (86.6%))		5.9 (5.2)
Eosinophils (found in 8 subjects (6.6%))		0.2 (0.5)
Ciliated cells (found in 66 subjects (54.5%))		0.7 (1.2)

* Hyposmic: Screening 12 Test score < 8, ** performed on 121 subjects, PNIF: peak nasal inspiratory flow, SNOT-22: Sino-Nasal Outcome-22 Test.

**Table 5 ijerph-17-07075-t005:** Main study results.

Independent	Dependent	Sub-Population	Means Difference	*ρ* Coefficient	*p*-Value
**PNIF**					
	FEV1	-	-	0.2	**0.03**
**Smoking Status**					
	Tiffeneau Index	-	−4.25	-	**0.0007**
	PNIF	-	3.12	-	0.72
	FEV1	-	0.25	-	0.12
	FEV1%	-	−1.75	-	0.51
	FeNO	-	−4.32	-	0.32
	SNOT-22	-	6.76	-	**0.003**
	Screening 12 Test	-	−0.02	-	0.95
	VAS for nasal obstruction	-	0.90	-	**0.02**
	Number of Neutrophils at NC	-	−0.006	-	0.60
**Dust concentration**					
	Tiffeneau Index	-	−3.2	-	**0.017**
		Smokers	−4.2	-	**0.021**
		Non-smokers	0.04	-	0.54
	FEV1	-	−0.15	-	0.35
	FEV1%	-	−0.92	-	0.74
	FeNO	-	−8.60	-	0.06
	SNOT-22	-	0.41	-	0.86
	Screening 12 Test	-	0.41	-	0.25
	VAS for nasal obstruction	-	0.24	-	0.56
	Number of Neutrophils at NC	-	−1.39	-	0.17
**Exposure time**					
	Tiffeneau Index	-	-	−0.12	**0.02**
		Smokers	-	−0.20	**0.002**
		Non-Smokers	-	0.04	0.54
	Number of Neutrophils at NC	-	-	0.15	**0.03**
	PNIF	-	-	0.13	0.73
	FEV1	-	-	0.16	0.48
	FEV1%	-	-	0.003	0.81
	FeNO	-	-	0.22	0.22
	SNOT-22	-	-	−0.09	0.36
	Screening 12 Test	-	-	−0.01	0.36
	Vas for nasal Obstruction	-	-	−0.02	0.11

PNIF: peak nasal inspiratory flow; SNOT-22: Sino-Nasal Outcome-22 Test; FEV1: forced expiratory volume in the 1st second; Tiffeneau index: FEV1/FVC (FVC: forced vital capacity); FeNO: fractional exhaled nitric oxide; NC: nasal cytology; VAS: Visual Analog Scale. Significance in bold.

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
