# Peer review of "Upper and Lower Respiratory Signs and Symptoms in Workers Occupationally Exposed to Flour Dust"

_ijerph, 2020, doi:10.3390/ijerph17197075_

Round 1

Reviewer 1 Report

Occupational allergic disease remains a significant issue in bakeries, so any study in this area is useful, to confirm or challenge finding from previously published work. The authors have undertaken a significant cross-sectional study  of bakers in a particular area of Italy as part of a surveillance programme.  They have collected symptoms data and spirometry related to the lower respiratory tract, skin prick test data against a considerable array of allergenic material, together with symptoms and signs related to upper respiratory/nasal. 

However in my opinion there are significant deficits in the way some of the data is, or is not, presented to the reader, the nature of statistical analysis utilised and thus the conclusions drawn from their study.  I also found some of table confusing in exactly knowing what information is being presented; readers will not have had the immersion in the date like the researchers. My comments below are identified by line numbers.

  1. line 51 and elsewher. "Atopic". Atopy means diferent things to different people. To my occupational colleagues it means a subject is  sensitised to more than one common environmental allergens in reference to them being sensitised to occupational allergens.  Could the authors ensure make it clear of what they mean and ensure consistency wherever it is used. I nowdays tend to think of subjects showing mulitple sensitisation rather than being mono-sensitised.
  2. line 58 alpha amylase. I presume this is added or exogenuous amylase?
  3. lines 75-77. Could authors use the convention of ony using acronyms after being fully described. E.g. SNOT-22 and VAS are only explained in lines 106-107. 
  4. Lines 79, 87-88, 139. I can find no description of the questionnaire used for lower respiratory tract symptoms and how it was employed to minimise the issues the authors note in lines 313-315. Were work-related symptoms collected. In line 139 10% reported what lower respiratory symptoms. There is a deficit in this paper concerning the collection and analysis of lower respiratory symptoms, as presented it means nothing.
  5. line 83. I would suggest here is the place for description of the various bakeries artisan, industrial. 
  6. line101. CECA71?
  7. line 122-125. How was the  sampling done? I gather they are personal samples. Are the samples considered representative of each workers usual job?
  8. line 126-132. Parametric analyses done but with no indication that highly skewed data was logged or other transformations used (e.g. table 1 exposure time appears highly skewed.  Simple univariate statistical approach used which does not allow consideration of which of possible mutiple independent explanatory factors are significant e.g. which factors  (smoking status, dust concentration, duration of exposure) are independently significant in influencing FEV1/FVC.
  9. line 134/135. explanantion of which Italian law and what is meant to be done by employer/employee. 
  10. Table 1. at top you have  mean (sd) "overall age" and then by sex. As currently laid out not immediately obvious to the reader.  Again for smoking I presume mean(sd) column refers to age of the subgroups, but it is no immediately obvious. 
  11. Line 141. " atopy familiarity". Do the authors mean report a family history of atopy or multiple sensitisations to common environmental allergens?
  12. Line 139.  What lower respiratory symptoms; suggestive of being work-related?
  13. Table 2.  See earlier comments on "atopy" Would it be better to rearrange the table so the individual flour allergens are after the overall flor allergen figure, as the mite allergens are. Is there an overall figure for those allergens at the bottom of the table that could be considered "environmental".  What constitutes "Airborne allergens" at the top of part B. 
  14. Line 148. "Flour mites", table 2 " storage mites". What differences do the authors imply? In  the context of the UK Derm. pteronys. is not considered a storage mite, but a house dust mite. 
  15. Table 2 line 156-156. What analysis rather simple description of the SPT data is given?  For example, is being  positive SPT to one of more environmental allergens a risk factor for being positive to the flour allergens? effects of smoking status  on positivity?
  16. line 150-156 and table 3. Analysis of data of spirometry and FeNO data. Was positivity for SPT in flour, mite or environmental allergens  a significant risk factor for bronchial obstruction or reduction of pulmonary volumes? Was abnormality of  FeNO significantly related to one or more positive SPT test? Any analysis of abnormality of spirometry and smoking status, duration of occuational exposure or dust levels?
  17. Line 153. Is (23.7+/- 25.0 range 2-166) the normal reference range as suggested by the layout or the distribution in the cohort?
  18. Line 167 "glasses" do the authors mean "slide"?
  19. Table 4. could the authors check that the layout is comperhensible to readers. For example the Visual Analoge Scale has data under the neading N(%).... is this right? After all  you cannot have 0.3 of a worker! Is this mean(sd) of the VAS scores? Similaary for the Screening 12 results and Nasal Cytologywhat are the numbers?.  Nasal Cytology not Citology.
  20. Line 173 not sure that dust levels to 3 decimal figure are warranted.
  21. Table 5 data.  Need a satistical  analysis  to see if smoking status, dust concentraion or duration of exposure are significan,t independent risk  factors for the changes in the Tiffenau Index, not three single univariate analyses. 
  22. Do figures 5 and 6 add anything to the paper? 
  23. Line 226-227 would prefer this is the statistical outcome of a more sophistcated approach than suggested in table 5. 
  24. Line 240. How does this statement about mites tally with lines 63-64?
  25. Lines 241-243. While I agree, the reality is that, certainly in the UK, the diagnosis of the majority of work-related asthma is done without inhalation challenges. 
  26. Lines 245-246. Is this noted in the results?
  27. Lines 247- 251. Fundamentally  the authors seem to accept the argument of epitope cross-reactivity between allergens without consideing that  multiple sensitisation may possibly indicate an increased genetic susceptibility to sensitisation in the occupational setting.
  28. Lines 312-315. What precautions did the authors take to reduce the influence of bias in questionnaire completion, 
  29.  Line 322. Surely table 1 suggests that the mean of exposure was 14 years, not that all workers had been necessarily  exposed for many years >14 years. 
  30. I note that the authors in lines 323-325 do allow for the possibility of a "healthy worker effect". What are the limitationsof a cross-sectional study in the outcomes highlighted?

Author Response

Open Review

English language and style

( ) Extensive editing of English language and style required
( ) Moderate English changes required
(x) English language and style are fine/minor spell check required
( ) I don't feel qualified to judge about the English language and style

Yes

Can be improved

Must be improved

Not applicable

Does the introduction provide sufficient background and include all relevant references?

(x)

( )

( )

( )

Is the research design appropriate?

(x)

( )

( )

( )

Are the methods adequately described?

( )

( )

(x)

( )

Are the results clearly presented?

( )

( )

(x)

( )

Are the conclusions supported by the results?

( )

(x)

( )

( )

Comments and Suggestions for Authors

Occupational allergic disease remains a significant issue in bakeries, so any study in this area is useful, to confirm or challenge finding from previously published work. The authors have undertaken a significant cross-sectional study  of bakers in a particular area of Italy as part of a surveillance programme.  They have collected symptoms data and spirometry related to the lower respiratory tract, skin prick test data against a considerable array of allergenic material, together with symptoms and signs related to upper respiratory/nasal. 

However in my opinion there are significant deficits in the way some of the data is, or is not, presented to the reader, the nature of statistical analysis utilised and thus the conclusions drawn from their study.  I also found some of table confusing in exactly knowing what information is being presented; readers will not have had the immersion in the date like the researchers. My comments below are identified by line numbers.

We are grateful to the reviewer for the suggestions that will surely make the manuscript better. Point by point:

  1. line 51 and elsewhere. "Atopic". Atopy means diferent things to different people. To my occupational colleagues it means a subject is  sensitised to more than one common environmental allergens in reference to them being sensitised to occupational allergens.  Could the authors ensure make it clear of what they mean and ensure consistency wherever it is used. I nowdays tend to think of subjects showing mulitple sensitisation rather than being mono-sensitized.

Reply: Atopy refers to the genetic tendency to develop allergic diseases such as allergic rhinitis, asthma and atopic dermatitis (eczema). Atopy is typically associated with a high immune response to common allergens, particularly inhaled allergens and food allergens, regardless of monosensitization or polysensitization and sensitization to common or occupational allergens. This sentence has been added in introduction section (lines 54-57)

  1. line 58 alpha amylase. I presume this is added or exogenous amylase?

Reply: Alpha amylase is an enzyme that hydrolyses starch and is added to flour to help increase the volume of bread preparation (see lines 62-63).

  1. lines 75-77. Could authors use the convention of ony using acronyms after being fully described. E.g. SNOT-22 and VAS are only explained in lines 106-107.

Reply: As suggested, we have fully entered the meaning of some acronyms that were missing. 

  1. Lines 79, 87-88, 139. I can find no description of the questionnaire used for lower respiratory tract symptoms and how it was employed to minimise the issues the authors note in lines 313-315. Were work-related symptoms collected. In line 139 10% reported what lower respiratory symptoms. There is a deficit in this paper concerning the collection and analysis of lower respiratory symptoms, as presented it means nothing.

Reply: we thank the reviewer for his/her observation. Lower respiratory symptoms have been collected by means of the Asthma Control Test (ACT) questionnaire. Subjects have been considered having lower respiratory symptoms when the scores were lower than 20. Materials/Methods and Result section have been modified (112, 156-157). Table 3 has been modified reporting ACT result.

  1. line 83. I would suggest here is the place for description of the various bakeries artisan, industrial. 

Reply: the distinction was added in the text (lines 90-93).

  1. line101. CECA71?

Reply: explained in the text (lines 114-115).

  1. line 122-125. How was the sampling done? I gather they are personal samples. Are the samples considered representative of each workers usual job?

Reply: we have added in the text the indications for a better comprehension of the samplers (lines 138-139).

  1. line 126-132. Parametric analyses done but with no indication that highly skewed data was logged or other transformations used (e.g. table 1 exposure time appears highly skewed.  Simple univariate statistical approach used which does not allow consideration of which of possible mutiple independent explanatory factors are significant e.g. which factors  (smoking status, dust concentration, duration of exposure) are independently significant in influencing FEV1/FVC.

Reply: The skewness has been analyzed and considered in the modelling data; however, among all available variables only the exposure time shows a moderate Pearson moment coefficient of skewness G1=1.3. Analyzing all models, considering a log transformed exposure time, results do not dramatically change and the inferential conclusions are always the same. For this reason, we preferred to consider the original scale variable in order to facilitate interpretations. We have now modified the text of the results (sections 3.6 and 3.7) adding also this information. Multiple regression models have been implemented for all relevant variables (FEV1, FEV1%, Tiffeneau, PNIF, FeNO, nasal cytology, VAS, Screening 12, SNOT 22, etc.), but no relevant results were obtained. The only multiple interactions that have been observed are those outlined in Table 5 between Tiffeneau Index and smoke behavior in dust concentration and exposure time. In addition, multivariate analysis (factor analysis and correlation analysis) has been implemented, taking all relevant variables together. No significant or useful results have been found, so we have not added them in the paper.

We have now included in the Statistical analysis (materials and methods section) a more detailed specification of what has been done.

  1. line 134/135. explanantion of which Italian law and what is meant to be done by employer/employee.

Reply: We thank the reviewer for the observation. We believe that the indication of the specific Italian low is too specific for this contest, so the sentence has been deleted.

  1. Table 1. at top you have  mean (sd) "overall age" and then by sex. As currently laid out not immediately obvious to the reader.  Again for smoking I presume mean(sd) column refers to age of the subgroups, but it is no immediately obvious. 

Reply: We agree with the reviewer and modified Table 1 following his/her suggestions. Now it should be clearer.

  1. Line 141. " atopy familiarity". Do the authors mean report a family history of atopy or multiple sensitisations to common environmental allergens?

Reply: The text describes the clinical characteristics of the study population. The clinical history divides the subjects into those with a history of familial atopy and those with a history of multiple sensitization to aero-allergens. Differentiation is important because they are two independent clinical aspects.

  1. Line 139.  What lower respiratory symptoms; suggestive of being work-related?

Reply: according to ACT questionnaire, the principal lower respiratory symptoms considered were: wheezing, cough, breathlessness, waking-up during the night for tight chest. The symptoms have been added to the text (line 163-164).

  1. Table 2.  See earlier comments on "atopy" Would it be better to rearrange the table so the individual flour allergens are after the overall flor allergen figure, as the mite allergens are. Is there an overall figure for those allergens at the bottom of the table that could be considered "environmental".  What constitutes "Airborne allergens" at the top of part B.

Reply: we are grateful to the reviewer for the for giving us the opportunity to review the table and improve it by dividing the categories of aero-allergens.

  1. Line 148. "Flour mites", table 2 " storage mites". What differences do the authors imply? In  the context of the UK Derm. pteronys. is not considered a storage mite, but a house dust mite.

Reply: Derm. pteronys. can be included in house dust mite. After table 2 modification, the list of allergens is now clearer (see modified table 2).

  1. Table 2 line 156-156. What analysis rather simple description of the SPT data is given?  For example, is being  positive SPT to one of more environmental allergens a risk factor for being positive to the flour allergens? effects of smoking status  on positivity?

Reply: We analyzed the relationship between flour allergens and environmental allergens prick tests positivity by finding no significant relationships. In addition, we did not find any effect of smoking or demographic characteristics on flour allergens. We added a sentence by making explicit these analyses.

  1. line 150-156 and table 3. Analysis of data of spirometry and FeNO data. Was positivity for SPT in flour, mite or environmental allergens  a significant risk factor for bronchial obstruction or reduction of pulmonary volumes? Was abnormality of  FeNO significantly related to one or more positive SPT test? Any analysis of abnormality of spirometry and smoking status, duration of occuational exposure or dust levels?

Reply: We thank the reviewer for his/her suggestion. We did the suggested statistical analysis finding a significant relationship (see Results, Section 3.1).

  1. Line 153. Is (23.7+/- 25.0 range 2-166) the normal reference range as suggested by the layout or the distribution in the cohort?

Reply: As the referee pointed out, the mean, sd and range refer to the observed data and were in the wrong position. We decided to delete the parenthesis since the same information is given in Table 3.

  1. Line 167 "glasses" do the authors mean "slide"?

Reply: thanks for observation. We change the text as suggested (line 203).

  1. Table 4. could the authors check that the layout is comperhensible to readers. For example the Visual Analoge Scale has data under the neading N(%).... is this right? After all  you cannot have 0.3 of a worker! Is this mean(sd) of the VAS scores? Similaary for the Screening 12 results and Nasal Cytology what are the numbers?.  Nasal Cytology not Citology.

Reply: We apologize. It was a mistake.  According to reviewer suggestions, we modified Table 4.

  1. Line 173 not sure that dust levels to 3 decimal figure are warranted.

Reply: we are grateful to the reviewer for the comment. Gravimetric determination of the dust was carried out by weighting the filters before and after sampling using a XPR6UD5 microbalance with the detection threshold of 0.0005 mg (Mettler Toledo, Columbus, OH).

  1. Table 5 data.  Need a satistical  analysis  to see if smoking status, dust concentraion or duration of exposure are significan,t independent risk  factors for the changes in the Tiffenau Index, not three single univariate analyses. 

Reply: A detailed multiple analysis (including regression models and additive models) has been implemented to identify significant relationships between all the available variables (including not only smoking status, dust concentration and duration of exposure, but also demographical quantities, other clinical parameters, and prick tests’ results). The only significant finding is the joint effect of smoking status and dust concentration as shown in Table 5. As mentioned before, in Materials and Methods (Statistical analyses section) we now included a more detailed specification of the multiple statistical analysis that was done.

  1. Do figures 5 and 6 add anything to the paper? 

Reply: According to the reviewer suggestion, both figures have been deleted. Figures’ numbers have been modified accordingly

  1. Line 226-227 would prefer this is the statistical outcome of a more sophistcated approach than suggested in table 5. 

Reply: As said above, our statistical approach was quite sophisticated and detailed, including different multiple regression models. However, since we only found a small number of significant results, we preferred to present those in a simpler way, as in Table 5, rather than presenting the model parameters estimations with significance and other relevant quantities.

  1. Line 240. How does this statement about mites tally with lines 63-64?

Reply: We thank the reviewer for his/her comment. We highlighted the difference of our study results with the data reported in literature (Discussion, lines 284-288).

  1. Lines 241-243. While I agree, the reality is that, certainly in the UK, the diagnosis of the majority of work-related asthma is done without inhalation challenges. 

Reply: We agree with the reviewer that the diagnosis of the majority of work-related asthma is done without inhalation challenges. Nevertheless, official guidelines recommend specific inhalation challenge for the diagnosis of occupational asthma and occupational rhinitis.

  1. Lines 245-246. Is this noted in the results?

Reply: the following sentence has been added to the results section (lines: 183-189): “Bakers sensitized to cereal flours showed an average exposure time of 16.15 (SD=10.2) years, while those not sensitized showed an average exposure time of 13.72 (SD=12.2) years”. In the Discussion the sentence has been modified.

  1. Lines 247- 251. Fundamentally  the authors seem to accept the argument of epitope cross-reactivity between allergens without consideing that  multiple sensitisation may possibly indicate an increased genetic susceptibility to sensitisation in the occupational setting.

Reply: We have not referred to the genetic susceptibility of sensitization to occupational aeroallergens in polysensitized workers because the analysis of the pattern of sensitization has shown a sensitization to aeroallergens of working origin in 53% of the subjects (storage mites and flours) without parallel sensitization to common aeroallergens (house dust mite and grass pollen).

  1. Lines 312-315. What precautions did the authors take to reduce the influence of bias in questionnaire completion

Reply: Scientifically validated international questionnaires have been used also for the collection of pulmonary symptoms (ACT). This has been specified in material and methods and the questionnaire results are reported in table 3.

  1.  Line 322. Surely table 1 suggests that the mean of exposure was 14 years, not that all workers had been necessarily  exposed for many years >14 years. 

Reply: thank you for your suggestion: we have changed the text accordingly (line 369).

  1. I note that the authors in lines 323-325 do allow for the possibility of a "healthy worker effect". What are the limitations of a cross-sectional study in the outcomes highlighted?

Reply: We agree with the reviewer suggestion and added in the Discussion section a final sentence recalling the possible bias of the healthy worker effect (lines 371-373).

Reviewer 2 Report

I have carefully read the manuscript titled "Upper and Lower Respiratory Signs and Symptoms in Workers Occupationally Exposed to Flour Dusts". The paper addresses an interesting issue concerning the possible correlations between higher/lower respiratory signs/symptoms, inflammation biomarkers and occupational exposure to flour dust. It appears well organized and written in an accurate form. I can express a positive opinion for its publication in IJERPH.

As a minor comment, the authors should check that all the abbreviations used in the text are detailed in their extensive form the first time they are cited in the text.

Author Response

Open Review

English language and style

( ) Extensive editing of English language and style required
( ) Moderate English changes required
(x) English language and style are fine/minor spell check required
( ) I don't feel qualified to judge about the English language and style

Yes

Can be improved

Must be improved

Not applicable

Does the introduction provide sufficient background and include all relevant references?

(x)

( )

( )

( )

Is the research design appropriate?

(x)

( )

( )

( )

Are the methods adequately described?

(x)

( )

( )

( )

Are the results clearly presented?

(x)

( )

( )

( )

Are the conclusions supported by the results?

(x)

( )

( )

( )

Comments and Suggestions for Authors

I have carefully read the manuscript titled "Upper and Lower Respiratory Signs and Symptoms in Workers Occupationally Exposed to Flour Dusts". The paper addresses an interesting issue concerning the possible correlations between higher/lower respiratory signs/symptoms, inflammation biomarkers and occupational exposure to flour dust. It appears well organized and written in an accurate form. I can express a positive opinion for its publication in IJERPH.

As a minor comment, the authors should check that all the abbreviations used in the text are detailed in their extensive form the first time they are cited in the text.

Reply: We are grateful for the reviewer's appreciation of our work. As suggested, we have fully entered the meaning of some acronyms that were missing.

Reviewer 3 Report

The manuscript consisted of 142 bakers to study respiratory signs/symptoms, inflammation biomarkers, and occupational exposure to flour dust. Comprehensive measurements were implemented, including skin prick tests (SPT), spirometric lung function tests, peak nasal inspiratory flow (PNIF), nasal symptoms scales with nasal cytology, and sino-nasal quality of life, Visual Analog Scales (VAS) for the main chronic rhinosinusitis symptoms, fractional exhaled nitric oxide (FeNO), the Screening 12 test along with the SNOT-22 questionnaire, and an environmental sampling of dust were performed in all bakeries. Finally, they concluded that long-term exposure to bakery dusts can lead to a status of minimal nasal inflammation and allergy.

I have some questions,

The study included 26 bakeries in the province of Padua. How many bakeries in this province? Were the studied bakeries randomly or intentionally selected?

In these 26 bakeries, how many bakers totally? Were the 142 bakers randomly or intentionally selected?

If not randomly selected, then it should be discussed the limitations, then the first and second paragraphs of discussion should re-write.

Skin prick test results showed high prevalence of atopy status, allergic 55%, should be discussed and comparison with other literature.

Figure 2. Correlation between FEV1 and PNIF, I suggested to change the FEV1 to FEV1 %pred. I mean to show the correlation between FEV1%pred and PNIF.

Author Response

Open Review

English language and style

( ) Extensive editing of English language and style required
( ) Moderate English changes required
( ) English language and style are fine/minor spell check required
(x) I don't feel qualified to judge about the English language and style

Yes

Can be improved

Must be improved

Not applicable

Does the introduction provide sufficient background and include all relevant references?

( )

(x)

( )

( )

Is the research design appropriate?

( )

(x)

( )

( )

Are the methods adequately described?

( )

(x)

( )

( )

Are the results clearly presented?

( )

(x)

( )

( )

Are the conclusions supported by the results?

( )

(x)

( )

( )

Comments and Suggestions for Authors

The manuscript consisted of 142 bakers to study respiratory signs/symptoms, inflammation biomarkers, and occupational exposure to flour dust. Comprehensive measurements were implemented, including skin prick tests (SPT), spirometric lung function tests, peak nasal inspiratory flow (PNIF), nasal symptoms scales with nasal cytology, and sino-nasal quality of life, Visual Analog Scales (VAS) for the main chronic rhinosinusitis symptoms, fractional exhaled nitric oxide (FeNO), the Screening 12 test along with the SNOT-22 questionnaire, and an environmental sampling of dust were performed in all bakeries. Finally, they concluded that long-term exposure to bakery dusts can lead to a status of minimal nasal inflammation and allergy.

I have some questions,

We are grateful to the reviewer for the suggestions that will surely make the manuscript better. Point by point:

The study included 26 bakeries in the province of Padua. How many bakeries in this province? Were the studied bakeries randomly or intentionally selected?

Reply: as added in the text, the bakeries were included randomly. The total number of bakeries in Padua Province was added in the text (lines 89-92).

In these 26 bakeries, how many bakers totally? Were the 142 bakers randomly or intentionally selected?

Reply: see above 

If not randomly selected, then it should be discussed the limitations, then the first and second paragraphs of discussion should re-write.

Reply: being bakeries randomly selected, we believe that the first and the second paragraphs are appropriate.

Skin prick test results showed high prevalence of atopy status, allergic 55%, should be discussed and comparison with other literature.

Reply: No many data are reported in literature. To the best of our knowledge, in general population, atopy prevalence has been reported to be about 20%. This data has been added and commented in the Discussion section (lines 284-288). 2 references have been added and the reference list has been changed accordingly.

Figure 2. Correlation between FEV1 and PNIF, I suggested to change the FEV1 to FEV1 %pred. I mean to show the correlation between FEV1%pred and PNIF.

Reply: We thank the reviewer for his/her comment. Anyway, giving to the fact that PNIF is an individual measured nasal parameter, we believe that it is better to compare PNIF with an individual measured pulmonary parameter more than a predicted one. For this reason, the figure has not been changed and the correlation between FEV1 and PNIF is presented.

Round 2

Reviewer 3 Report

I have no more comments.

Author Response

We would like to thank the reviewer for the comment and for the time that he spent to review the manuscript. We think his comments helped us to improve it.

Best regards

Andrea Trevisan